# OpenReview forum: "White-box Multimodal Jailbreaks Against Large Vision-Language Models"
_acmmm.org/ACMMM/2024/Conference — MM2024 Poster_

### Official Review · Reviewer_MmyJ · 2024-05-06

**Rating:** 4
**Confidence:** 4

**Summary:**

This paper attempts to guide LVMs output towards disturbing and offensive content through malicious inputs. It constructs adversarial image prefixes and adversarial text suffixes through a two-stage optimization strategy: Embedding toxic semantics into adversarial image prefix and text-image mltimodal optimization for maximizing affirmative response probability. This process forms a set of Universal Master Key (UMK). The effectiveness of the method is experimentally verified.

**Strengths:**

* Studying the security and privacy of LVMs is a meaningful issue.
* The combination of adversarial image prefixes and text suffixes is a clever approach. Reasonably leveraging multimodal attacks enhances the success rate of the attacks.
* The comparative method is quite representative. Introducing the image-image attack strategy provides more interesting insights.

**Limitations:**

* In most cases, the white-box scenario is still not the primary target for attackers. Building perturbations with transferability may have more practical significance.
* There are also some white-box attack methods designed for small-scale multimodal pre-trained models, such as [1], [2], and more advanced white-box attack methods built upon these approaches. If the authors can explain the advantages relative to these methods, I am willing to increase my score.
* I noticed that in Algorithm 1, the attack is divided into two steps. In the first step, it searches for an image perturbation that encourages the model to output text from the set S. In the second step, it searches for a text perturbation that encourages the model to output the target text t. Why isn't the loss function in the first step optimized to encourage the model to output text t, forming a progressive relationship between the two optimizations? Would this be more effective?

[1] Towards Adversarial Attack on Vision-Language Pre-training Models
[2] AdvCLIP: Downstream-agnostic Adversarial Examples in Multimodal Contrastive Learning

**Suitability:**

3

---

### Official Review · Reviewer_Gca3 · 2024-05-15

**Rating:** 3
**Confidence:** 3

**Summary:**

The authors' goal is to find a universal pair of image/text input that is effective in letting Large Vision-Language models (VLMs) generate harmful outputs. The authors combined previous methods, VAJM (effective in producing adversarial images) and GCG (effective in producing adversarial text), to generate a pair of image/text input that maximizes the probability of the model generating undesirable output.

**Strengths:**

- The proposed method is effective in generating harmful behaviors against VLMs, resulting in high success rate against targeted models.
- The authors included informative ablation studies, showing the necessity of adding toxic semantics into the image.

**Limitations:**

- While the method is definitely effective, it is a combination of two existing methods, which is not very novel.
- While the ablation studies indicated that the step of adding toxic semantics into the image is necessary, some explanation of why this is the case can be beneficial. It is an observation that certain images that induce the model to generate unwanted outputs with empty texts are also effective with non-empty texts. Some references that confirm this assumption will be beneficial in the analysis of the ablation studies.

**Suitability:**

3

---

### Official Review · Reviewer_eSAH · 2024-05-19

**Rating:** 4
**Confidence:** 2

**Summary:**

This paper presents a novel white-box multimodal jailbreak attack against Large Vision-Language Models (VLMs). The authors propose a comprehensive strategy that simultaneously attacks both text and image modalities, exploiting a broader range of vulnerabilities within VLMs. The methodology includes a dual optimization objective aimed at generating highly toxic affirmative responses. Specifically, the attack involves optimizing an adversarial image prefix to generate harmful content without text input and then jointly optimizing this prefix with an adversarial text suffix to maximize affirmative responses to malicious queries. The combined adversarial image prefix and text suffix are termed the Universal Master Key (UMK). Experimental results demonstrate the effectiveness of this approach, achieving a 96% success rate in jailbreaking MiniGPT-4.

**Strengths:**

1. The concept behind this work is intriguing as it addresses a gap in current research, which mainly concentrates on single-mode attacks.

2. The authors conduct extensive experiments across multiple datasets, including Advbench and RealToxicityPrompts, to validate their approach. The results consistently show that the proposed UMK outperforms existing unimodal attack methods in terms of attack success rate.

3. The paper is well-structured, with clear explanations of the methodology and detailed descriptions of the experiments.

**Limitations:**

1. While the proposed method shows high effectiveness on MiniGPT-4, its transferability to other VLMs is not thoroughly explored. Given the differences in model architectures, parameters, and tokenizers across various VLMs, further investigation into the method's applicability to other models would enhance the paper's comprehensiveness and practical relevance.

2. The primary focus of the paper is on the attack methodology and its success. However, the paper provides limited discussion on potential defense mechanisms to counteract these multimodal attacks. Since the ultimate goal of identifying these vulnerabilities is to guide the enhancement of model security, a more in-depth exploration of defensive strategies would be beneficial.

3. Expression issues
a) line 17 "affirmative responses with high toxicity" -> "highly toxic affirmative responses"
b) line 583 "use a single A100 GPU with 80GB of memory in all experiments" -> "using a single A100 GPU with 80GB of memory for all experiments."

**Suitability:**

3

---

### Official Review · Reviewer_pMUm · 2024-05-25

**Rating:** 4
**Confidence:** 3

**Summary:**

This article presents a novel multimodal attack strategy for large multimodal white-box models. By optimizing the attack to generate harmful content and optimizing the response to users’ malicious requests, the multimodal attack proposed in this paper shows a significant improvement over single-modality attacks on the MiniGPT-4 model.

**Strengths:**

1. This paper proposes a novel multimodal attack strategy for large multimodal white-box lmodels. Compared to unimodal attacks, the multimodal attacks presented in this paper demonstrate more significant effects.
2. The paper optimizes the attacks through two objectives, effectively addressing two issues in attacking large models: (a) providing affirmative responses to malicious user content without generating toxic content, and (b) generating toxic content that is unrelated to the user's instructions.
3. The paper adopts a stricter criterion for measuring the success of an attack, considering an attack successful only when the model's output is both harmful and relevant to the user's instructions. This stricter criterion ensures a more rigorous evaluation.
4. Comprehensive ablation experiments were conducted, fully demonstrating the effectiveness of the proposed method.

**Limitations:**

1. Compared to other works on attacking large models, this paper uses a different criterion for determining attack success, employing manual evaluation. Although the paper's method for determining attack success is more reasonable, the large amount of harmful content generated by the model means the manual evaluation results may not be suitable for public release. To enhance the persuasiveness of the attack effects, it may be necessary to use keyword or semantic-based attack success criteria similar to those in other papers. It would be helpful to see the performance evaluated using the same methods as other papers for ASR results.
2. This paper focuses on responses to harmful textual instructions, but since the attacked model is multimodal, many harmful instructions could be embedded in image data (e.g., an image containing content that instructs the model to provide a bomb-making tutorial) or the image content could be related to harmful instructions (e.g., inferring the location or personal information of a person in a photo and providing targeted criminal actions). It is unclear whether the attack method in this paper can effectively generate harmful responses to such harmful instructions.

**Suitability:**

3

---

### Meta-Review · Area_Chair_ZpzK · 2024-07-01

**Recommendation:** Accept (Poster)
**Confidence:** 5

**Metareview:**

Initially, the paper received three borderline accept and one borderline reject ratings. After the rebuttal, all four reviewers increased their scores and recommended acceptance of the paper. I recommend accepting this paper as a poster for the conference. In addition to the potential flaws mentioned in the paper, the authors should also consider the potential issues raised by Reviewer pMUm (in Final Rating Justification).